# Dysfunction in IGF2R Pathway and Associated Perturbations in Autophagy and WNT Processes in Beckwith–Wiedemann Syndrome Cell Lines

**DOI:** 10.3390/ijms25073586

**Published:** 2024-03-22

**Authors:** Silvana Pileggi, Elisa A. Colombo, Silvia Ancona, Roberto Quadri, Clara Bernardelli, Patrizia Colapietro, Michela Taiana, Laura Fontana, Monica Miozzo, Elena Lesma, Silvia M. Sirchia

**Affiliations:** 1Medical Genetics, Department of Health Sciences, Università degli Studi di Milano, 20142 Milan, Italy; silvana.pileggi@unimi.it (S.P.);; 2Pharmacology, Department of Health Sciences, Università degli Studi di Milano, 20142 Milan, Italyelena.lesma@unimi.it (E.L.); 3Department of Biosciences, Università degli Studi di Milano, 20133 Milan, Italy; 4Medical Genetics, Department of Pathophysiology and Transplantation, Università degli Studi di Milano, 20122 Milan, Italy; 5Dino Ferrari Centre, Neuroscience Section, Department of Pathophysiology and Transplantation, Università degli Studi di Milano, 20122 Milan, Italy; 6Unit of Medical Genetics, ASST Santi Paolo e Carlo, 20142 Milan, Italy

**Keywords:** Beckwith–Wiedemann Syndrome (BWS), IGF2R, *IGF2*, imprinting, autophagy, WNT pathway

## Abstract

Beckwith–Wiedemann Syndrome (BWS) is an imprinting disorder characterized by overgrowth, stemming from various genetic and epigenetic changes. This study delves into the role of *IGF2* upregulation in BWS, focusing on insulin-like growth factor pathways, which are poorly known in this syndrome. We examined the IGF2R, the primary receptor of IGF2, WNT, and autophagy/lysosomal pathways in BWS patient-derived lymphoblastoid cell lines, showing different genetic and epigenetic defects. The findings reveal a decreased expression and mislocalization of IGF2R protein, suggesting receptor dysfunction. Additionally, our results point to a dysregulation in the AKT/GSK-3/mTOR pathway, along with imbalances in autophagy and the WNT pathway. In conclusion, BWS cells, regardless of the genetic/epigenetic profiles, are characterized by alteration of the IGF2R pathway that is associated with the perturbation of the autophagy and lysosome processes. These alterations seem to be a key point of the molecular pathogenesis of BWS and potentially contribute to BWS’s characteristic overgrowth and cancer susceptibility. Our study also uncovers alterations in the WNT pathway across all BWS cell lines, consistent with its role in growth regulation and cancer development.

## 1. Introduction

Beckwith–Wiedemann syndrome (BWS, OMIM #130650) is an overgrowth disorder characterized by variable major features such as macrosomia, macroglossia, abnormal wall defects, and embryonal tumors (i.e., Wilms tumor, hepatoblastoma, neuroblastoma, and rhabdomyosarcoma). Complications like prematurity, hypoglycemia, cardiomyopathy, macroglossia, or tumor development can lead to early mortality [1].

BWS is an imprinting disorder associated with genetic and epigenetic defects affecting imprinted growth regulatory genes, *IGF2/H19* and *CDKN1C/KCNQ1OT1*, that are located on chromosome 11p15.5 and independently regulated through methylation of two imprinting control regions (IC1 and IC2) [2].

Approximately 60–70% of BWS cases result from loss of methylation (LoM) at IC2 on the maternal chromosome. Mosaic paternal uniparental disomy (patUPD) involving duplication of the paternally derived 11p15.5 without maternal contribution occurs in 20–25% of cases. Gain of methylation (GoM) at IC1 on the maternal chromosome is observed in 5–10% of patients. Germline mutations in the maternally expressed *CDKN1C* gene are found in 5% of sporadic BWS and ~40% of cases with a positive family history, while chromosomal rearrangements (translocations or inversions) are relatively rare (~1% of cases) [3,4,5,6,7,8,9].

Increased Insulin-like Growth Factor 2 (*IGF2*) gene expression, a consequence of imprinting abnormalities, is observed in BWS, while its reduction correlates with Silver–Russell syndrome (SRS), another imprinting disorder caused by defects on the same chromosome region, but with the opposite growth phenotype [10,11]. These observations evidence the critical role of IGF2 regulation in disorders characterized by growth abnormalities.

*IGF2*, encoding for a ubiquitous growth factor, regulates pre- and postnatal growth and development. IGF2 is a secreted protein influencing a variety of cellular processes, including metabolism, proliferation, survival, and differentiation [12,13,14]. It interacts with multiple receptors, notably, Insulin-like Growth Factor 1 Receptor (IGF1R), Insulin Receptor (INSR), and Insulin-like Growth Factor 2 Receptor (IGF2R) [14]. The binding to the IGF1R activates the phosphoinositide-3-kinase (PI3K)-AKT pathway, crucial for cell growth, differentiation, and specific gene expressions [15]. However, the primary receptor of IGF2 is IGF2R, a 300 kDa membrane-bound glycoprotein, which primarily participates in the transport of lysosomal enzymes from the trans-Golgi apparatus via early and late endosomes for their subsequent internalization in lysosomes [16,17,18,19,20]. This non-signaling receptor is involved in several physiological processes and, through the sequester of IGF2, is able to prevent the accumulation of excessive and deleterious levels of IGF2, especially during embryonic development [15]. Perturbation in IGF2/IGF2R signaling has been associated with autophagy impairment in several human diseases (such as Parkinson’s disease and cancer) [21,22] and similar evidence has been observed in mice brains after treatment with IGF2 [23].

Autophagy is a tightly regulated catabolic process for self-degradation of cellular components that are engulfed in autophagosomes that subsequently fuse with lysosomes for the digestion of the luminal cargo. At physiological levels, autophagy is indispensable to maintain the normal homeostasis and metabolism of tissues and to allow cellular adaptation to external conditions [24,25]. Thus, this process is induced by cellular stress and is required to preserve cell fitness through the recognition of autophagy-selective substrates by specific receptors [26,27]. Altered autophagy by genetic or acquired defects has been associated with human pathologies, such as neoplastic, cardiometabolic, inflammatory, and degenerative diseases [24].

Among the several pathways involved in developmental processes, a crosstalk between the signaling mediated by WNT/β-catenin and autophagy has been well established, both directly associated with cellular homeostasis, maintenance, and differentiation [28].

WNT signaling is involved in important cellular processes during both embryonic development and adult life. Its dysregulation can lead to a broad range of growth aberrations and pathologies, including cancer, which are typical clinical features of BWS patients [29,30].

Given the importance of IGF for cell growth, survival, autophagy [31], and migration, the maintenance of correct IGF2 levels is crucial in normal growth and development [32]. Despite that, the impairment of these pathways in BWS syndrome is poorly known, and the role and the specific targets induced by IGF2 receptor activation are not completely clear.

The study aims to investigate the IGF2 signaling in lymphoblastoid cell lines (LCLs) from BWS patients with different genetic/epigenetic defects and from healthy controls. We focus on the signaling pathways of IGF receptors, investigating IGF2R expression and localization, and PI3K/AKT and mitogen-activated protein kinase (MAPK)/extracellular signal-regulated kinase (ERK) pathways. In addition, starting from the recent evidence of the direct role of IGF2 and IGF2R in autophagy [3], we also explore the autophagy/lysosomal cascade in BWS LCLs, examining the following key factors: Unc-51-like kinase 1 (ULK1) and Beclin-1 and their phosphorylated forms (P-ULK and P-Beclin), whose recruitment and activation are essential steps of the autophagic process.

Finally, we also assessed the expression profiles of a panel of genes associated with the WNT and imprinting pathways, which are fundamental in growth control.

## 2. Results

For our experiments, we employed lymphoblastoid cell lines obtained from BWS patients with different genetic and epigenetic defects (IC1 GoM, IC2 LoM, or UPD) and healthy pediatric controls previously characterized in our laboratory [33]. The cell line derived from an SRS patient, which has the opposite defect to BWS, was also included, as an additional control [33]. Detailed information about the epigenetic status of the 11p15 imprinted region of LCLs used in the study and a schematic overview of the experimental design are summarized in Table 1.

### 2.1. Imprinted and Imprinted-Related Gene Expression in BWS and Control Cell Lines

Given that BWS is an imprinting disorder characterized by growth defects, we investigated the expression profiles of a panel of 17 genes associated with the imprinting and growth pathways in BWS and control LCLs using the Nanostring technique. Out of these 17 genes, 11 were expressed in the LCLs (Table 2). Notably, among them, only *IGF2R*, coding for the primary receptor of IGF2, was differentially expressed (differentially expressed gene, DEG) in patients’ cell lines and in particular was upregulated with an unadjusted *p*-value of 0.0454.

### 2.2. IGF2R Expression and Localization Analyses in BWS and Control Cell Lines

IGF2R, a membrane glycoprotein, plays a crucial role in capturing and internalizing IGF2. Additionally, it has a cytoplasmic localization due to its involvement in lysosomal protein trafficking from the trans-Golgi network to lysosomes [16,35]. To assess IGFR2 localization and expression in our cell lines, we performed immunofluorescence and Western blot analyses. For immunofluorescence, we used primary antibodies against IGF2R and LAMP1, a lysosomal-associated membrane protein 1 [36], crucial for lysosomes function [37]. As shown in Figure 1A, IGF2R localization in control cell lines is noted on the plasma membrane and in a perinuclear region, likely representing the Golgi apparatus. Differently, BWS LCLs (Figure 1B) exhibit an almost absent membrane-associated signal and a reduced perinuclear signal, particularly in BWS IC1. The BWS-UPD cell line displays two distinct signal patterns, suggesting the presence of two cellular subpopulations with varying IGF2R expression/localization, herein named BWS UPD A and BWS UPD B (Figure 1C). The BWS UPD A subpopulation represents the majority of the cells (about 75%) and displays an IGF2R signal pattern similar to controls, while the BWS UPD B subpopulation (about 25–30%) seems to mirror the other BWS cell lines. This heterogeneity can be explained by the known mosaicism in patients with patUPD of chromosome 11 [38]. Accordingly, our BWS-UPD cell line displayed mosaicism with about 30% of cells with UPD, as confirmed by SNP array. Figure 1A,B show a reduction in LAMP1 signal in BWS cell lines compared to controls. Notably, Figure 1A reveals a LAMP1 and IGF2R colocalization (yellow signal) in controls, predominantly in CTRL1 and CTRL2 cell lines; this feature is lost in BWS IC1 and IC2 lines (Figure 1B). Western blot analysis strengthens the immunofluorescence results, showing a lower IGF2R expression in BWS LCLs compared to control and SRS cell lines (Figure 1D).

Further analysis on IGF2R distribution by using a line scan of IGF2R fluorescence intensity for each cell line (Figure 2A) demonstrated IGF2R misbehavior in BWS LCLs. Unlike controls, where IGFR2 is mainly on the plasma membrane, in BWS cells, it is less expressed and internalized, adjacent to the nucleus, presumably within the Golgi apparatus. To deeply assess the receptor distribution, we used a method previously applied by our group [39] that evaluates the distance between the geometrical center of the cell (centroid) and the center of fluorescence mass signal, as a measure of fluorescence distribution. The distance between centroid and fluorescence mass center indicates the degree of polarization [40,41]: the shorter the distance, the more uniform the fluorescent signal, whereas higher values are indicative of polarized fluorescence. By this approach, we confirmed a mislocalization of IGF2R in BWS LCLs, particularly a significant polarized fluorescence signal in the BWS IC1 (ANOVA test = 0.082, *t*-test 0.017), BWS IC2 (ANOVA test < 0.0001, *t*-test 0.00023), and BWS UPD B (ANOVA test < 0.0001, *t*-test < 0.0001) LCLs compared to controls (Figure 2B). No substantial differences were observed among the entire BWS-UPD population, BWS UPD A subpopulation, and controls, which showed a similar immunofluorescence pattern, whereas a significant difference was observed between the BWS UPD A and BWS UPD B subpopulation (ANOVA test < 0.0001, *t*-test < 0.0001).

### 2.3. Analysis of IGF2R Targets in BWS and Control Cell Lines

In light of the diverse cellular signaling pathways controlled by the IGF2R activation [42], we evaluated potential targets by using a Profiling Phosphotyrosine Signaling array (PathScan^®^ Signaling Array Kit, Cell Signalling Technologies, Danvers, MA, USA). The preliminary results indicated the involvement of the AKT pathway, particularly a reduction in AKT phosphorylation at Thr 308 and, mainly, at Ser 473 (Appendix A). The decreased AKT phosphorylation (Ser 473) was further confirmed through Western blot analysis, showing lower levels in BWS IC1 and IC2 LCLs compared to CTRL, SRS, and BWS UPD LCLs (Figure 3A and Appendix A). As observed for AKT activation, the phosphorylation of S6 at Ser235/236, a downstream target of the PI3K/AKT/mammalian target of the Rapamycin complex 1 (mTORC1) pathway [43], was also reduced in BWS IC1 and IC2 LCLs compared to the controls, SRS, and BWS UPD LCLs. The expression levels of both AKT and S6 remained similar across all the LCL groups. Additionally, the expression and phosphorylation at Thr202/Tyr204 of ERK1/2 (p44 and p42 MAP Kinase) did not exhibit variation across all LCL samples (Figure 3A and Appendix A), although a slight difference was observed in the PathScan^®^ Signaling Array which analyzed MEK1/2 (ERK1/2 regulator) phosphorylation at Ser221 and Ser217/221 (Appendix A).

To deepen the AKT impairment, we evaluated the activation of glycogen synthase kinase 3 (GSK-3), as frequently phosphorylated by AKT. GSK-3 is a serine/threonine protein kinase and a component of the PI3K/PTEN/AKT/GSK-3/mTORC1 pathway [44]. Notably, the phosphorylation levels of GSK at Ser21 (GSK-3α) and at Ser9 (GSK-3β) were much lower in BWS IC1 and IC2 LCLs compared to CTRL, SRS, and BWS UPD LCLs (Figure 3B and Appendix A). Additionally, the phosphorylation of cyclic adenosine monophosphate (cAMP) response element-binding protein (CREB), regulated by GSK-3β via AKT and mitogen-activated protein kinases (MAPKs), showed a slight inhibition in BWS IC1 and IC2 LCLs (Figure 3B and Appendix A). CREB and GSK-3 levels were comparable in all groups.

The observed differences in the phosphorylated targets between BWS-IC1/IC2 LCLs and BWS-UPD LCLs might be due to the mosaic condition characterizing the BWS UPD cell line, as previously noted (Figure 1C). Conversely, the similar activation pattern between controls and BWS-UPD LCLs could be explained by the presence of only 25–30% of cells with UPD in the latter.

In summary, these findings suggest a dysregulation of the PI3K/AKT/mTORC1 pathway in BWS LCLs.

### 2.4. Analysis of Autophagy in BWS LCLs

In light of the IGF2R modifications in BWS cells and the known involvement of this receptor in lysosomal activity and autophagic function, we investigated the abundance and the activation levels of key autophagic proteins. Consistently with the reduced activation of the AKT-mTORC1 pathway, the inhibitory phosphorylation of Unc-51-like kinase 1 (ULK1), an enzyme required to initiate autophagy, was lower in BWS LCLs than in CTRL and SRS LCLs, suggesting an aberrant activation of the autophagic process (Figure 4 up). In BWS LCLs, the phosphorylation of Ser30 of Beclin1, a key protein in autophagy, was also reduced compared to CTRL LCLs (Figure 4 down). These data suggest impaired autophagic activity in BWS LCLs compared to CTR LCLs.

### 2.5. WNT Pathway Analysis in BWS and Control Cell Lines

Mutations in key genes of the WNT pathway, such as *CTNNB1* and *AXIN1*, have been observed in BWS embryonal tumors, including hepatoblastoma, Wilms tumor, and pancreatoblastoma [45]. In addition, alterations in the WNT pathway have been associated with growth defects, a hallmark feature of BWS [28]. To investigate the possible involvement of WNT pathway alteration in BWS, we evaluated the expression of 180 genes of the WNT pathways using the Nanostring approach. Of these, 163 were expressed in our LCLs (Appendix A) [46].

Our analysis identified 29 DEGs with an unadjusted *p*-value < 0.05 between BWS and control samples; 17 were upregulated and 12 were downregulated (Figure 5A and Table 3). Among the 29 DEGs, 17 reached a Benjamini–Hochberg (BH) adjusted *p*-value < 0.1. Notably, the *TP53* gene was the most upregulated (unadjusted *p*-value of 2.81 × 10^−5^; BH *p*-value of 0.00446), while *PLAUR*, *PRKCB*, and *DKK4* were the most downregulated (unadjusted *p*-value of 0.000288, 0.000798, and 0.00102, and BH *p*-value of 0.0229, 0.0377, and 0.0377, respectively).

Interestingly, among the differentially expressed genes, *AXIN1*, *CTNNB1*, *MAKP10,* and *DKK4* have previously been reported to mutate or deregulate in tumors of BWS patients [45,47].

Principal Component Analysis (PCA), an unsupervised pattern recognition analysis allowing an easy visualization of expression differences between samples, was performed using the nSolver software (version 4.0). The PCA revealed a clustered distribution of BWS patients distinct from the scattered distribution of controls, suggesting that the alteration of the WNT pathways is a common condition in BWS (Figure 5B).

Pathway enrichment analysis is a bioinformatic technique used to analyze gene expression data aimed at identifying altered biological pathways or networks in a set of experimental data. This analysis was performed using the nSolver software and highlighted that the most altered sub-pathways belonging to the Vantage 3DTM RNA WNT Pathways Panel in BWS cells were the canonical WNT and the transcription factor pathways (Figure 5C, left panel). These findings are further displayed in the box plots presented in Figure 5C, right panel. These results highlight the involvement of the WNT pathways in BWS pathogenesis. In particular, Figure 5D provides a schematic representation of the observed expression alterations in the three main WNT signaling pathways belonging to the Vantage 3DTM RNA WNT Pathways Panel (specific annotation of genes is reported in Appendix A and Table 3). The altered nodes of these pathways, as identified by Pathview (nSolver Advanced Analysis Software 4.0; Figure 5D) were p53 (DEG: *TP53*), Frizzled (DEG: *FZD2*), WNT (DEG: *WNT10A*), GBP (DEG: *FRAT1*), JNK (DEGs: *MAPK9* and *MAPK10*), BAMBI (DEG: *BAMBI*), DKK (DEG: *DKK4*), and cycD (DEG: *CCND1*).

## 3. Discussion

Beckwith–Wiedemann syndrome is an imprinting disorder characterized by overgrowth and predisposition to embryonal tumors such as Wilms tumors, hepatoblastoma, neuroblastoma, and rhabdomyosarcoma [1]. This syndrome shows high genetic/epigenetic heterogeneity involving alterations in the 11p15.5 region, which harbors four imprinted genes: *IGF2* and *H19* (IC1 locus), and *CDKN1C* and *KCNQ1OT1* (IC2 locus) [2]. *IGF2,* encoding an embryonic growth factor, is involved in different processes including survival, proliferation, differentiation, autophagy, and tumorigenesis [12,13,14,48,49] and its expression is predominantly from the paternal allele in most adult tissues [50,51].

IGF2R, as the primary receptor of IGF2, serves as a repository for the IGF2 growth factor, targeting it to lysosomes for degradation and thereby controlling its concentration. IGF2R impinges on essential processes in different types of cells, in an opposite way to IGF2 [52]; it is thus recognized as a tumor suppressor due to its role in clearing IGF2 [53].

The complex relationship between IGF2 and IGF2R has been explored in various pathologies. Alberini and coworkers have demonstrated that, through IGF2R binding, the administration of IGF2 promotes autophagy via endosomal/lysosomal activities; this activation is associated with new protein synthesis [23,54,55,56,57]. They hypothesize an IGF2R-mediated balance between protein synthesis and degradation, suggesting that the receptor may promote genesis or mobilization of endosomes, which could serve as platforms for de novo mRNA translation [58].

The expression of IGF2 and its receptor is interrelated: IGF2 overexpression correlates with abnormal growth and increased cell proliferation [59], while IGF2R overexpression leads to the opposite phenotype, as demonstrated in studies on mice in which IGF2R was overexpressed in smaller animals, whereas its absence or deficiency was associated with overgrowth [16].

Scalia et al. recently reviewed the diverse pathomechanisms behind loss of imprinting which lead to IGF2 dysregulation in BWS and consequent bi-allelic IGF2 expression [60]. Contrastingly, Silver–Russell syndrome (SRS), a disease associated with the opposite genetic defect, is characterized by reduced growth and lower expression of IGF2 [61]. Nevertheless, the impairment of pathways related to IGF signaling in BWS remains poorly known.

Our study on IGF2R expression and localization in BWS-derived cell lines reveals IGF2R mRNA overexpression, coupled with reduction and mislocalization of the receptor. Indeed, in BWS cells, IGF2R is localized near the nucleus, most likely in the Golgi apparatus, whereas in controls it is mainly on the plasma membrane. Moreover, BWS cells show, according to the distance analysis between centroid and center of mass, a dishomogeneous IGF2R distribution, compared to controls. It is possible that IGF2 levels are key to the level of activation of its receptor and therefore to the changes that we observed intracellularly.

The discrepancy we found in IGF2R mRNA and protein expression in BWS cells might be explained by a rapid IGF2R turnover due to an increase in IGF2, typical of the syndrome, which is a hypothesis supported by Alberini’s findings regarding the increase in the protein turnover triggered by IGF2 supply [62,63].

IGF2 acts through its binding to various IGF/insulin receptors, with the highest affinity to IGF2R, exerting autocrine, paracrine, and endocrine effects [64]. The interaction between IGF2 and IGF2R regulates different biochemical pathways associated with pathological processes whose mechanisms are not yet completely understood. By exploring downstream targets of IGF2R in BWS LCLs, our analysis reveals a downregulation in the AKT/mTOR signaling axis, characterized by reduced phosphorylation of AKT and S6.

PI3K/AKT signaling has several downstream targets, among which GSK-3, when phosphorylated on Ser 9 residue by AKT, is inhibited in its activity. The proteins phosphorylated by GSK-3 are mainly inactivated by targeting for proteasome degradation and/or might change their subcellular localization, altering the physiological activation [65]. In BWS IC1 and IC2 LCLs, consistent with the weak AKT and S6 phosphorylation, GSK-3 phosphorylation at Ser 9 (GSK-3β) and Ser21 (GSK-3a) is lower than in controls, suggesting an impaired regulation of GSK-3. Of note, GSK-3 expression can affect different biochemical processes in tumorigenesis and can modulate cellular senescence, cell cycle arrest, apoptosis, and chemoresistance [66]. Considering the large number of substrates that GSK-3 can have, we focused on the transcription factor CREB for its role in regulating cellular proliferation [67]. CREB phosphorylation is slightly inhibited in BWS LCLs compared to controls, suggesting a predominant impairment of the AKT/GSK–3/mTOR axis. AKT/GSK-3/mTOR signaling is known to be involved in controlling the incorporation of receptors (in particular IGF1R) into the membrane [68]. Therefore, a decrease in the activation of this axis could decrease the IGF2 receptor amount in the membrane and lead to more important changes in intracellular signaling. Finally, despite ERK1/2 hyperactivation via IGF2R being described in a BWS mouse model [69], in our LCLs, ERK1/2 phosphorylation did not differ between the control, BWS, and SRS cell lines.

GSK-3 is also a critical component of the WNT signaling pathway, whose principal function is cell–cell communication through the regulation of cell proliferation, differentiation, and migration, as well as apoptosis [29]. WNT is one of the best-known evolutionary conserved pathways in embryonic development. It is therefore easy to imagine what effects dysregulation of this pathway may have on pre- and post-natal growth. Analyzing the expression profiles of a panel of WNT-associated genes in our cell lines, we found a dysregulation of this pathway in BWS cells. Interestingly, several genes with altered expression are involved in development, differentiation, and migration, such as *MMP9*, *BMP4*, and *T* genes (the most upregulated genes), and *GDNF* and *CCND1* (the most downregulated genes).

Altogether, our results suggest an alteration of the WNT and autophagy pathways in BWS mediated by aberrant expression and localization of IGF2R. Given its role in tumorigenesis, the receptor abnormalities emerging from our data may contribute to the increased risk of developing embryonic cancer that characterizes BWS patients.

Alterations of the IGF2/IGF2R signaling are also associated with perturbations of autophagy in different disorders [21,22,23,50,54]. In addition, in cancer cell lines, a dysregulation of autophagic function and reduced degradative capacity of lysosomes following IGF2R knockdown has been observed [35].

Our findings confirm the interplay among IGF2R, autophagy, and lysosomal processes. We observed reduced LAMP1 signal, in particular in BWS cell lines with IC1 and IC2 defects, alongside decreased phosphorylation of ULK1 and Beclin, key enzymes in the autophagic process. This suggests impaired autophagic/lysosomal activity in BWS cells. Of note, autophagy is partially controlled by GSK-3 and mTORC1 [70], which we found dysregulated in BWS LCLs.

Interestingly, our BWS cell lines show similar behaviors, regardless of IC1 or IC2 defects. The observed IGF2R alterations in BWS IC1 and BWS UPD, where the IGF2 locus is directly involved, are conceivable; however, it is not in BWS IC2, which presents defects in the second locus. Our previous work, by Rovina et al., demonstrated a direct crosstalk between the chromatin structure of the two imprinted regions, suggesting that defects in one locus could lead to abnormalities in the other [33]. This may explain the IGF2R alterations also observed in BWS IC2 cells. The crosstalk between the two imprinted regions is also supported by similar data obtained from all the BWS cell lines regarding autophagy/lysosome pathways and WNT expression profiles, which clearly identify BWS cell lines as a distinct group with respect to controls. Our findings indicate a high homogeneity of BWS cell lines, regardless of the genetic defect; only the UPD cell line is slightly different, showing less evident alterations, and this can be explained by the mosaic condition of this cell line.

In conclusion, the alteration of IGF2R observed in BWS cells is independent of the genetic/epigenetic defect and is associated with the perturbation of fundamental processes, such as the autophagy and lysosome pathways. These combined alterations seem pivotal in the molecular pathogenesis of BWS, contributing to hallmark features of the syndrome, such as overgrowth and cancer predisposition. Finally, the perturbation of the WNT pathway, a signaling axis involved in growth control and tumorigenesis, also emerges as a common feature in BWS cells.

This work only provides a picture of these pathways in immortalized cell lines and the observed alterations will have to be validated on fresh patient samples; however, this is the first evidence of the involvement of these pathways in Beckwith–Wiedemann syndrome and may provide the rationale for future studies.

## 4. Materials and Methods

### 4.1. Lymphoblastoid Cell Lines

The LCLs, already described by Rovina D et al. [33], are summarized in Table 1. Briefly, the LCLs were generated from three BWS patients with different genetic/epigenetic defects and four unaffected pediatric controls (CTRL 1–4). The study was approved by the Ethics Committee of Fondazione IRCCS Ca’ Granda Ospedale Maggiore Policlinico (no. 526/2015). Appropriate written informed consent was obtained from the patients’ parents. All the procedures performed in this study were in accordance with the 1964 Helsinki declaration and its later amendments.

BWS patient LCLs were established from patient blood samples, by Epstein–Barr virus transformation at the Galliera Genetic Bank (a member of the Telethon Network of Genetic Biobanks; project no. GTB12001).

Control and patient cell lines were cultured in RPMI 1640 medium supplemented with 10% heat-inactivated fetal bovine serum (Euroclone, Milan, Italy) and antibiotics (antibiotic-antimycotic 100×, Euroclone, Milan, Italy) at 37 °C in 5% CO_2_.

### 4.2. nCounter Analysis

Total RNAs were obtained using the Qiazol reagent (Qiagen, Hilden, Germany), followed by RNA purification by the RNeasy mini kit (Qiagen, Hilden, Germany), according to the manufacturer’s protocol. RNAs were eluted in 50 μL of RNase-free water. Concentration and purity were evaluated using Nanodrop (Thermo Fisher Scientific, Wilmington, DE, USA). RNA integrity was assessed with the Tape Station 2200 (Agilent, Santa Clara, CA, USA); RNA integrity number (RIN) values >7.0 were considered suitable for the experiments. Expression analysis was performed by nCounter using the Nanostring Vantage 3DTM RNA WNT Pathways Panel (Nanostring, Seattle, WA, USA), a panel including 180 genes associated with the WNT pathways and 12 reference genes for normalization (*CC2D1B*, *COG7*, *EDC3*, *GPATCH3*, *HDAC3*, *MTMR14*, *NUBP1*, *PRPF38A*, *SAP130*, *SF3A3*, *TLK2*, *ZC3H14*), customized with 17 imprinted and imprinted-related genes. The expression profiles were evaluated starting from 150 ng of total RNA for each sample.

We used Nanostring technology as it represents a medium-throughput platform to evaluate mRNA abundance profiles providing reproducible and fully automated analyses of the samples. The robustness of this technology is already validated in several papers [46,71,72]. The reliability of Nanostring technology is based on the ability to quantify the expression of multiple genes without amplification steps. Conversely, technical artifacts could be introduced in qPCR.

Nanostring data were analyzed by the nSolver Advanced Analysis Software 4.0 (NanoString, Seattle, WA, USA) considering a background threshold of 20 counts and excluding from the analysis all genes with counts above the threshold. Quality assessment was performed for each sample, and two quality control parameters common to all nCounter assays were considered: the Imaging QC that measures the percentage of the requested fields of view successfully scanned in each cartridge lane and the Binding Density QC that measures the reporter probe density on the cartridge surface in each sample lane. The Benjamini–Hochberg method was applied to reduce the false discovery rate (FDR), minimizing Type I errors (false positives). An unadjusted *p*-value ≤ 0.05 was considered significant.

### 4.3. Immunofluorescence Assay—Image Quantification and Statistics

The cells were cultured on glass slides, permeabilized with 70% methanol for 10 min, and dried in air. The primary antibodies against IGF2R (1:400; Cell Signalling Technologies, Danvers, MA, USA) and LAMP1 (1:600; Santa Cruz Biotechnology, Inc. Dallas, TX, USA) were applied overnight at 4 °C. The samples were incubated for 2 h at room temperature with Alexa 488- or Alexa 555-conjugated secondary antibodies (Molecular Probes, INC. Eugene, OR, USA). Nuclei were stained with DAPI (2 μg/mL; Sigma-Aldrich, Saint Louis, MO, USA). Images were acquired using a csu-w1 Nikon spinning disk confocal microscopy using a 100× objective or with an LEICA SP8 confocal microscope using a 40× objective with the sequential acquisition setting at four random fields in each sample. All images were processed with Fiji ImageJ analysis software (version 1.54).

The distribution of IGF2R was evaluated as previously described [39] by measuring the distance between the geometrical center of the cell (centroid) and the center of fluorescence mass (in this approach, more polarized fluorescent signals correspond to higher distance values). Remarkably, no difference in cell size was observed between controls and BWS cells, excluding the increased fluorescence distance between centroid and center of mass caused by an overall increase in cell dimensions.

Statistical tests were performed using Student’s *t*-test and ANOVA using GraphPad Prism 7.02.

### 4.4. Western Blot Analysis

Western Blot analysis was performed as previously described [73]. Briefly, cells were lysed in lysis buffer composed of 5 mM EDTA, 100 mM deoxycholic acid, and 3% sodium dodecyl sulphate and supplemented with protease inhibitors (Benzamidine 1 mM, PMSF 400 µM, Leupeptine 1 µg/mL, Aprotinin 10 µg/mL, Sigma-Aldrich, Saint Louis, MO, USA). Samples were boiled for 5 min, and 30 µg of proteins was loaded onto the gel with an appropriate concentration of acrylamide/bisacrylamide for SDS-PAGE and transferred to nitrocellulose membranes (Amersham, Arlington Height, IL, USA). After blocking at room temperature for 1 h with 5% dry milk (Merck, Darmstadt, Germany), membranes were incubated overnight at 4 °C with Cell Signalling Technologies (Danvers, MA, USA) antibodies against Phospho-AKT (Ser473) (1:1000; Cat. No. 4058), AKT (1:1000; Cat. No. 4685), phospho-S6 (Ser235/236) (1:1000; Cat. No. 2211), S6 ribosomal protein (1:1000; Cat. No. 2217), ULK1 (1:1000; Cat. No. 8054), Phospho-ULK1 (Ser757) (1:1000; Cat. No. 14202), Beclin1 (1:1000; Cat. No. 3459), Phospho-Beclin1 (S30) (1:100; Cat. No. 35955), IGF-2 Receptor (1:500; Cat. No. 14364), GSK3α/β (1:1000; Cat. No. 5676), Phospho-GSK3α/β (1:1000; Cat. No. 8566), CREB (1:1000; Cat. No. 4820), Phospho-CREB (Ser133) (1:1000; Cat. No. 9198), p44/42 MAPK (Erk1/2) (1:1000; Cat. No. 9102), Phospho-p44/42 MAPK (Erk1/2) (1:500; Cat. No. 9101) and β-actin (1:1000; Sigma-Aldrich, Saint Louis, MO, USA, Cat. No. A5441), The appropriate horseradish peroxidase conjugate-secondary antibodies (1:10,000; Thermo Scientific, Rockford, IL, USA, Cat. No. 31430/31460) were incubated for 1 h at room temperature, and the ECLT Prime Western Blotting System (Amersham, UK) or WesternBright Sirius HRP Substrate were used to reveal chemiluminescence. Images were acquired on a Kodak image station 1550 GL.

Densitometric analysis was performed using the Kodak MJ project program (Kodak, Milan, Italy) and the results were expressed as the mean value of phospho/total proteins for three independent experiments. Statistical analysis was performed with the GraphPad Prism 7.02 software (GraphPad Software, San Diego, CA, USA). *p*-values less than 0.05 were considered statistically significant in a two-way ANOVA. The results are presented as mean ± standard error of the mean (SEM).

### 4.5. Phospho-Array Profiler Analysis

Cells were lysed as described in the Western blot analysis section.

For the PathScan^®^ EGFR Signaling Antibody Array Kit (Chemiluminescent Readout) (Cell Signalling Technologies, Danvers, MA, USA, cat. No. 12622), 0.5 mg/mL of proteins was diluted in Array Diluent Buffer and analysis was performed according to the manufacturer’s instructions.

## Figures and Tables

**Figure 1 ijms-25-03586-f001:**
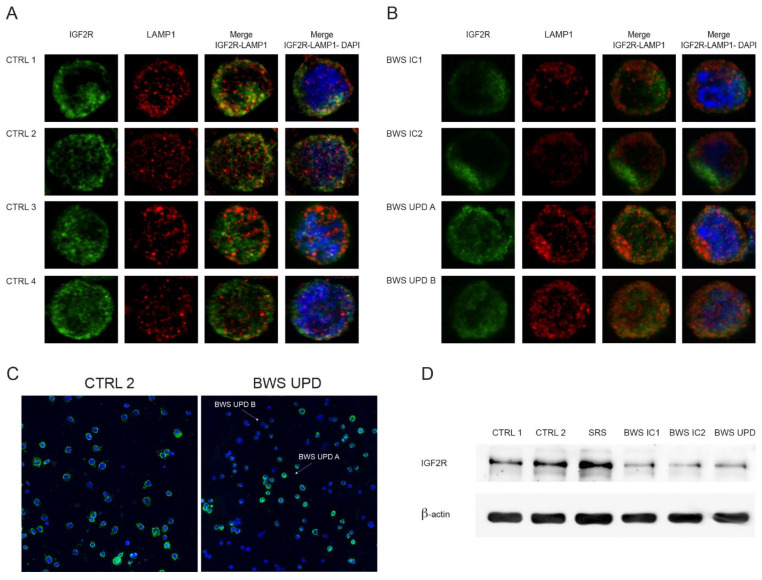
Localization and expression of IGF2R in control and BWS cell lines. (**A**,**B**) Immunofluorescence analysis in controls (**A**) and BWS (**B**) cell lines with a 100× objective using the antibodies against IGF2R (green signal) and lysosomal-associated membrane protein 1 (LAMP1) with a lysosomal marker protein (red signal). Nuclei were stained with DAPI (blue signal). Csu-W1 Nikon spinning disk confocal microscopy (**C**) Immunofluorescence analysis with a 40× objective using the antibodies against IGF2R (green signal), nuclei were stained with DAPI (blue signal) in CTRL2 and BWS UPD. Arrows indicate the two cellular subpopulations for UPD cells, UPD-A and UPD-B. LEICA SP8 confocal microscope. (**D**) IGF2R expression was evaluated by Western blot in CTRL, SRS, and BWS LCLs. Protein loading was normalized to β-actin and the shown images are representative of three independent experiments.

**Figure 2 ijms-25-03586-f002:**
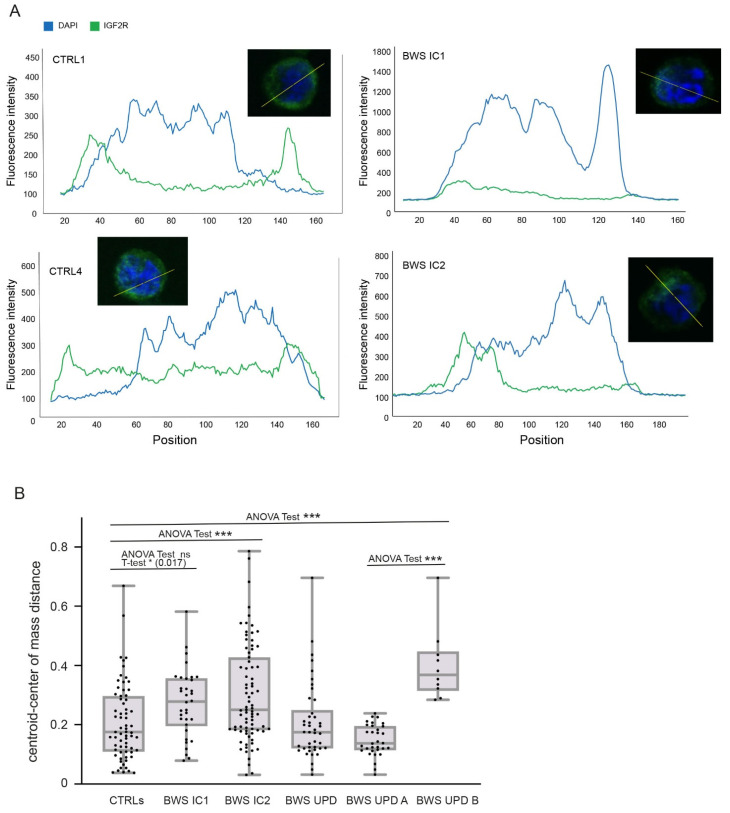
IGF2R distribution analysis. (**A**) Representative images of line scan of fluorescence intensity (yellow bars in figure’s miniatures) for CTRL 1 and CTRL 4, and BWS IC1 and BWS IC2 cell lines by using ImageJ software (version 1.54). The image of the analyzed cell is shown in miniature. (**B**) Graph shows the centroid and center of mass distance of almost 30 cells from 3 independent experiments for each cell line. Boxes include 50% of data points, lines represent the median distance, and whiskers report the minimum and maximum values. Differences (two-way ANOVA test and *t*-test) are indicated by asterisks (*** < 0.0001 and * < 0.05, respectively).

**Figure 3 ijms-25-03586-f003:**
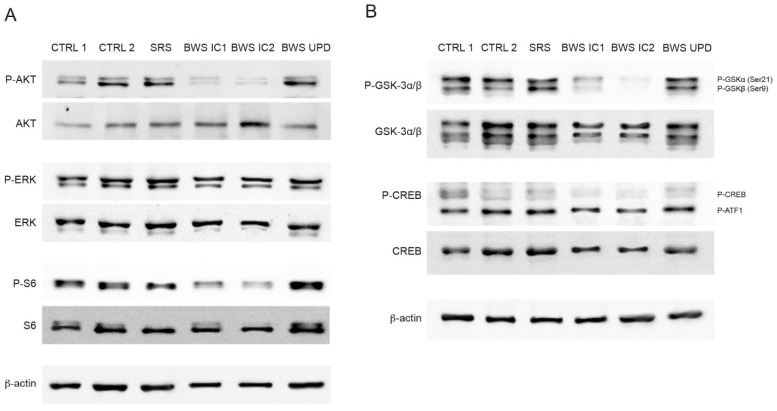
Analysis of possible targets of the IGF2R pathway in control and BWS cell lines. (**A**) Expression of the phosphorylation levels of AKT, ERK1/2, and S6 were analyzed by Western blot in controls (CTRL1 and CTRL2), SRS, and BWS cell lines. (**B**) Representative images of Western blot for phosphorylation and expression of GSK-3α/β and CREB are shown. In (**A**,**B**), protein loading was normalized to β-actin and the shown images are representative of three independent experiments.

**Figure 4 ijms-25-03586-f004:**
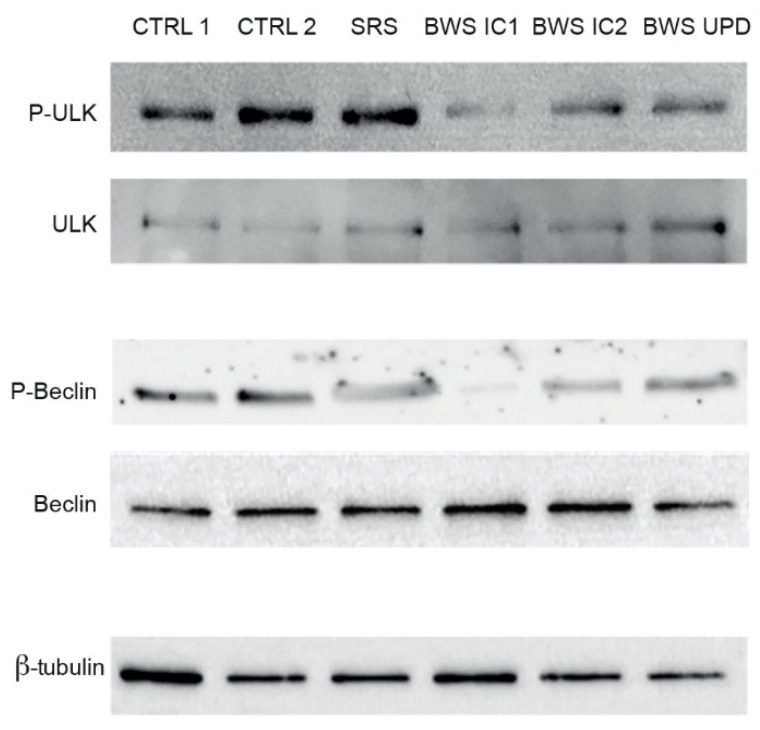
Evaluation of autophagy in control and BWS LCLs. Phosphorylation and expression of ULK and Beclin1 were analyzed by Western blot in control, SRS, and BWS cell lines. β-tubulin was used as loading control. The images are representative of three independent experiments.

**Figure 5 ijms-25-03586-f005:**
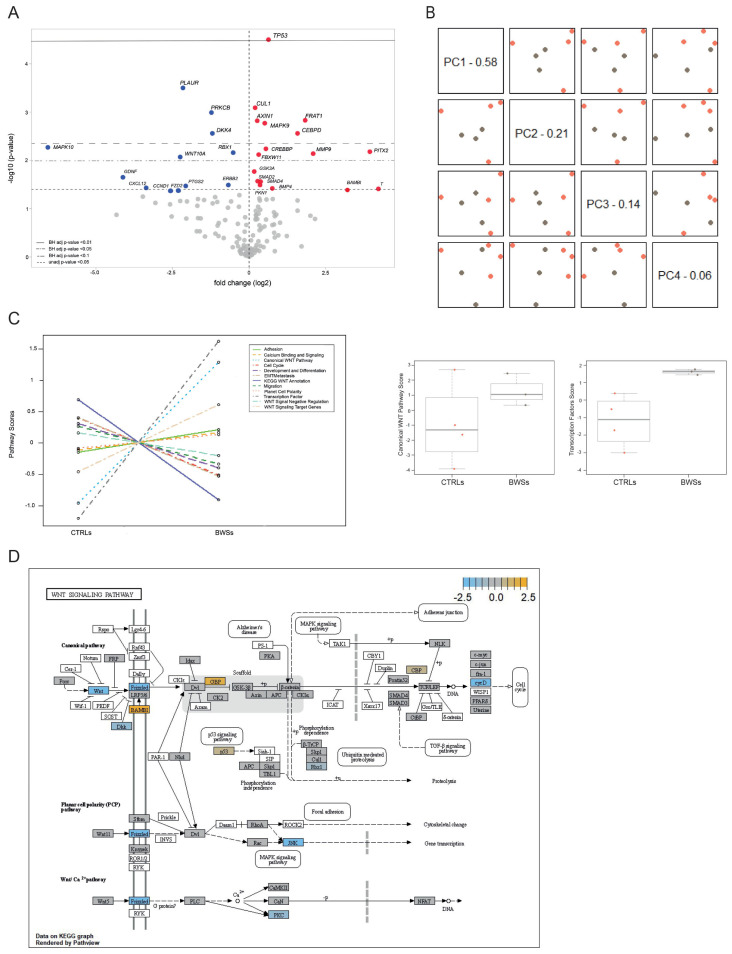
WNT pathway analysis in BWS and control cell lines. (**A**) Volcano plot of DEGs BWS compared to control cell lines. Upregulated genes are highlighted by red dots, while downregulated genes by blue dots. FDR Benjamini–Hochberg adjusted *p* values and unadjusted *p* value < 0.05 are indicated by horizontal lines. The VolcaNoser tool was used for creating volcano plots. (**B**) Principal Component Analysis distributed samples according to the first principal components in three BWS LCLs (gray dots) and four controls LCLs (orange dots). (**C**) Analysis of WNT panel’s sub-pathways. Left: trend plot of pathway scores vs. sample types (CTRLs and BWS). This image shows the differences of the expression of the genes belonging to the different sub-pathways of the Vantage 3DTM RNA WNT Pathways Panel between controls and BWS. Right: the analysis of the two most dysregulated sub-pathways (canonical WNT and the transcription factor) is depicted also as box plots. (**D**) Schematic representation of DEGs in the BWS cell lines belonging to the three main WNT pathways. Pathway nodes shown in white have no genes in the Vantage 3DTM RNA WNT Pathways Panel. Pathway nodes in gray have corresponding genes in the panel. However, no significant differential expression is observed. Nodes in blue and orange denote downregulation or upregulation in BWS compared to CTRLs. The nodes of the pathways that were found to be dysregulated by Pathview (nSolver Advanced Analysis Software 4.0) were p53 (DEG: *TP53*), Frizzle d (DEG: *FZD2*), WNT (DEG: *WNT10A*), GBP (DEG: *FRAT1*), JNK (DEGs: *MAPK9* and *MAPK10*), BAMBI (DEG: *BAMBI*), DKK (DEG: *DKK4*), and cycD (DEG: *CCND1*). PCA, pathway enrichment analysis, box plots, and schematic representation of DEGs were performed by nSolver software (Figures rendered by Pathview, nSolver Advanced Analysis Software 4.0).

**Table 1 ijms-25-03586-t001:** Epigenetic status of the 11p15 imprinted region of LCLs and schematic overview of the experimental design.

	LCLs Characterization				Study Design		
Cell Line	LCLs Methylation Level		LCLs Methylation Status	SNP Array	nCounter Analysis	Immunofluorescence	Western Blot
	IC1	IC2					
CTRL1	42%	43%	NM	NA	+	+	+
CTRL2	46%	44%	NM	NA	+	+	+
CTRL3	40%	40%	NM	NA	+	+	−
CTRL4	41%	42%	NM	NA	+	+	−
BWS-IC1	78%	42%	IC1 GOM	NA	+	+	+
BWS-IC2	44%	16%	IC2 LOM	NA	+	+	+
BWS-UPD	60%	27%	IC1 LOM/ IC2 GOM	30% UPD cells	+	+	+
SRS	28%	50%	IC1 GOM	NA	−	−	+

IC1 and IC2 methylation levels, obtained by pyrosequencing, in peripheral blood lymphocytes and lymphoblastoid cell lines of controls and BWS and SRS patients. Normal range: IC1 40–52%, IC2 39–50% [34]. NA: not analyzed. NM: normal methylation. GOM: gain of methylation. LOM: loss of methylation. +: performed analysis. −: not performed analysis.

**Table 2 ijms-25-03586-t002:** Quantitative expression analysis of the imprinted genes between BWS and control LCLs by nCounter Nanostring approach.

Gene	Accession	Log2 Fold Change	*p*-Value	BH *p*-Value
*IGF2R **	NM_000876.1:2605	0.487	0.0454	0.233
*PEG10*	NM_001040152.1:5000	−1.18	0.115	0.321
*MEST*	NM_177525.1:645	−2.58	0.128	0.339
*IGF1R*	NM_000875.4:4580	1.28	0.179	0.364
*GNAS-AS1*	NR_002785.2:1026	−0.482	0.227	0.415
*GNAS*	NM_080425.1:1910	0.127	0.294	0.502
*PLAGL1*	NM_006718.3:1872	0.226	0.435	0.652
*INSR*	NM_000208.2:525	−0.447	0.489	0.7
*KCNQ1OT1*	NR_002728.2:31875	−0.146	0.659	0.845
*IGF1*	NM_000618.3:491	−0.279	0.816	0.903
*FAM50B*	NM_012135.1:1272	0.0365	0.836	0.915

* Differentially expressed genes by nCounter analysis. Unadjusted *p*-value ≤ 0.05 was considered significant.

**Table 3 ijms-25-03586-t003:** Differentially expressed genes of the WNT pathway between BWS and control LCLs evaluated by nCounter Nanostring approach.

Gene	Accession	Log2 Fold Change	*p*-Value	BH *p*-Value	Pathway Annotation
*TP53*	NM_000546.2:1330	0.632	2.81 × 10^−5^	0.00446	KEGG WNT Annotation
*PLAUR*	NM_001005376.1:440	−2.16	0.000288	0.0229	Proteolysis
*CUL1*	NM_003592.2:1487	0.196	0.000798	0.0377	KEGG WNT Annotation
*PRKCB*	NM_212535.1:1750	−1.23	0.00102	0.0377	KEGG WNT Annotation
*FRAT1*	NM_005479.3:1100	1.83	0.00146	0.0377	Canonical WNT Pathway, KEGG WNT Annotation
*AXIN1*	NM_181050.1:135	0.265	0.00149	0.0377	Canonical WNT Pathway, KEGG WNT Annotation, WNT Signaling Negative Regulation
*MAPK9*	NM_139068.2:365	0.51	0.00166	0.0377	KEGG WNT Annotation
*CEBPD*	NM_005195.3:939	1.58	0.0027	0.0481	Transcription Factors
*DKK4*	NM_014420.2:640	−1.2	0.00272	0.0481	KEGG WNT Annotation
*MAPK10*	NM_002753.2:2080	−6.57	0.00526	0.079	KEGG WNT Annotation
*CREBBP*	NM_001079846.1:4818	0.547	0.00573	0.079	KEGG WNT Annotation
*PITX2*	NM_000325.5:1381	3.94	0.00649	0.079	Transcription Factors, WNT Signaling Target Genes
*RBX1*	NM_014248.2:162	−0.521	0.00682	0.079	KEGG WNT Annotation
*MMP9*	NM_004994.2:1530	2.09	0.00722	0.079	Calcium Binding and Signaling, Development and Differentiation, Migration, Proteolysis
*FBXW11*	NM_033645.2:3545	0.313	0.00745	0.079	KEGG WNT Annotation, WNT Signaling Negative Regulation
*WNT10A*	NM_025216.2:2255	−2.25	0.00837	0.0832	Calcium Binding and Signaling, Canonical WNT Pathway, KEGG WNT Annotation
*GSK3A*	NM_019884.2:480	0.167	0.0168	0.157	Canonical WNT Pathway
*GDNF*	NM_000514.2:580	−4.12	0.0222	0.185	Development and Differentiation, Migration
*SMAD2*	NM_005901.5:1678	0.278	0.0268	0.206	EMTMetastasis, KEGG WNT Annotation
*SMAD4*	NM_005359.3:1370	0.374	0.0272	0.206	KEGG WNT Annotation
*ERBB2*	NM_001005862.1:1255	−0.679	0.0319	0.222	EMTMetastasis
*PKN1*	NM_213560.1:2153	0.357	0.0322	0.222	EMTMetastasis
*PTGS2*	NM_000963.1:495	−2.07	0.0335	0.222	Calcium Binding and Signaling, Cell Cycle
*CXCL12*	NM_000609.5:210	−3.36	0.0367	0.226	EMTMetastasis
*BMP4*	NM_001202.3:395	0.757	0.0372	0.226	Development and Differentiation
*T*	NM_003181.2:1836	4.22	0.0383	0.226	Development and Differentiation, Transcription Factors
*BAMBI*	NM_012342.2:1010	3.2	0.0408	0.226	Canonical WNT Pathway
*CCND1*	NM_053056.2:690	−2.39	0.0425	0.226	Calcium Binding and Signaling, Cell Cycle, Development and Differentiation, KEGG WNT Annotation, WNT Signaling Negative Regulation, WNT Signaling Target Genes
*FZD2*	NM_001466.2:845	−2.36	0.0426	0.226	Calcium Binding and Signaling, Canonical WNT Pathway, KEGG WNT Annotation

## Data Availability

The data presented in this study are available on request from the corresponding author.

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
