# Peer review of "Dysfunction in IGF2R Pathway and Associated Perturbations in Autophagy and WNT Processes in Beckwith–Wiedemann Syndrome Cell Lines"

_ijms, 2024, doi:10.3390/ijms25073586_

Round 1

Reviewer 1 Report

Comments and Suggestions for Authors

The manuscript by Pileggi et al. examines the understudied role of the IGF2R pathway in Beckwith-Wiedemann Syndrome (BWS) cell lines. In general, the paper is well written and addresses an interesting and important topic with convincing data that utilizes Nanostring technology to detect gene expression, as well as phospho-array profiler analysis and bioinformatics.

Major points.

1.       As BWS is characterized by growth excess, are there any markers reflecting differences in cell proliferation between control and BWS cell lines?

2.       Figure 3: Although the legend says that Western blots are representative of 3 independent experiments, no quantification of protein bands or of phospho/total proteins ratio are presented. Based on the images shown, it looks as if no differences in p-S6,p-ERK and p-CREB would be observed as loading is not equal. The same may be true for Figure 4.

3.       Figure 5: As this manuscript is not specifically aimed at an audience very familiar with bioinformatic methods, much more explanation is required for the meaning of the results presented in this figure. Also, the quality of images should be improved. The characters, especially in shaded areas, are difficult to read.

4.       Figure 5E depicts genes of the WNT signaling pathway and indicates up- or down regulation in BWS cell lines and Table 3 presents all genes differentially expressed in BWS cell lines as compared to control cell lines. These results are derived from Nanostring analysis, but no confirmation is presented either by real-time PCR or by protein assessment to support the genomic results.

5.       No functionals test are performed to confirm that observed changes in these cell lines are the result of IGF1R downregulation, especially when considering that these cell lines are not original, but transformed.

Author Response

 As BWS is characterized by growth excess, are there any markers reflecting differences in cell proliferation between control and BWS cell lines?

Author response: Thank you for this comment. We know that our study does not investigate in depth some functional and cellular aspects. Our work aims to provide a picture of specific little-studied pathways in BWS, in immortalized cell lines derived from patients, hoping to give a valid rationale for future studies. In addition, the overgrowth aspects, typical of BWS, involve other tissues and not the hematopoietic cell lineage so lymphoblastoid cells are not the ideal cell substrate for cell biology in-depth studies.   

Figure 3: Although the legend says that Western blots are representative of 3 independent experiments, no quantification of protein bands or of phospho/total proteins ratio are presented. Based on the images shown, it looks as if no differences in p-S6,p-ERK and p-CREB would be observed as loading is not equal. The same may be true for Figure 4.

Author response: Thank you for this comment. As suggested, we quantified the phosphorylated forms of the proteins and the total protein bands. The results are shown in Supplemental Figure S2 as ratio of the relative intensity (Optical density: O.D.) of phospho/total proteins. The statistical analysis (ANOVA with Tukey test) of the phospho/total protein bands shows significant statistically differences only for some groups, however, the ratio trends indicated a reduction of the phosphorylated forms of P-AKT, P-S6, P-CREB, and P-GSK3a/β. 

Accordingly, we modified the Material and Methods section, “Western Blot Analysis” paragraph, adding the sentence “Densitometric analysis was performed by using Kodak MJ project program (Kodak, Milan, Italy) and results were expressed as the mean value of phospho/total proteins for three independent experiments. Statistical analysis was performed with the GraphPad Prism 7.02 software. (GraphPad Software, San Diego, CA, USA). p-values less than 0.05 were considered statistically significant in a two-way ANOVA. Results are presented as mean ± standard error of the mean (SEM).” (page 17, lines 517-522). 

We agree with the referee that ERK1/2 activation did not differ among the groups, as reported also in the manuscript (page 7 lines 201-205 and page 14 lines 374-375). 

Figure 5: As this manuscript is not specifically aimed at an audience very familiar with bioinformatic methods, much more explanation is required for the meaning of the results presented in this figure. Also, the quality of images should be improved. The characters, especially in shaded areas, are difficult to read.

Author response: thank you for your useful observations that allow us to better explain our results. To clarify this issue, we modified the results paragraph (page 8 from line 260 to line 279), Figure 5, and its legend. 

Results paragraph changes: “Principal Component Analysis (PCA), an unsupervised pattern recognition analysis  allowing an easy visualization of expression differences between samples, has been  performed using the nSolver software. PCA revealed a clustered distribution of BWS patients distinct from the scattered distribution of controls, suggesting that the alteration of the WNT pathways is a common condition in the BWS (Figure 5B). 

Pathway enrichment analysis is a bioinformatic technique used to analyze gene expression data aiming at identifying altered biological pathways or networks in a set of experimental data. This analysis was performed using nSolver software and highlighted  that the most altered sub-pathways belonging to the Vantage 3DTM RNA WNT Pathways Panel in BWS cells were the canonical WNT and the transcription factor pathways (Figure 5C, left panel). These findings are further displayed in the box plots presented in Figure 5C, right panel. These results highlight the involvement of the WNT pathways in BWS pathogenesis. In particular, Figure 5D provides a schematic representation of the observed expression alterations in the three main WNT signaling pathways belonging to the Vantage 3DTM RNA WNT Pathways Panel (specific annotation of genes is reported in Table S1 and Table 3). The altered nodes of these pathways, as identified by Pathview (nSolver Advanced Analysis Software 4.0; Figure 5D) were: p53 (DEG: TP53), Frizzled (DEG: FZD2), WNT (DEG: WNT10A), GBP (DEG: FRAT1), JNK (DEGs: MAPK9 and MAPK10), BAMBI (DEG: BAMBI), DKK (DEG: DKK4) and cycD (DEG: CCND1).” 

Figure 5 changes: we ensemble panel C and D because they are part of the same analysis and modified its legend as follows: “C) Analysis of WNT panel’s sub-pathways. Left: trend plot of pathway scores vs. sample types (CTRLs and BWS). This image shows the differences of the expression of the genes belonging to the different sub-pathways of the Vantage 3DTM RNA WNT Pathways Panel between controls and BWS. Right: the analysis of the two most dysregulated sub-pathways (canonical WNT and the transcription factor) is depicted also as box plots. D) Schematic representation of DEGs in the BWS cell lines belonging to the three main WNT pathways. Pathway nodes shown in white have no genes in the Vantage 3DTM RNA WNT Pathways Panel. Pathway nodes in gray have corresponding genes in the panel, however no significant differential expression is observed. Nodes in blue and orange denote downregulation or upregulation in BWS compared to CTRLs. The nodes of the pathways that were found to be dysregulated by Pathview (nSolver Advanced Analysis Software 4.0) were: p53 (DEG: TP53), Frizzled (DEG: FZD2), WNT (DEG: WNT10A), GBP (DEG: FRAT1), JNK (DEGs: MAPK9 and MAPK10), BAMBI (DEG: BAMBI), DKK (DEG: DKK4) and cycD (DEG: CCND1).

PCA, pathway enrichment analysis, box plots, and schematic representation of DEGs were performed by nSolver software (Figures rendered by Pathview, nSolver Advanced Analysis Software 4.0)”  

Finally, we are sorry that this figure did not appear clear but perhaps it lost resolution when loaded inside the manuscript. We have however submitted the high-resolution figures in the journal’s system. 

Figure 5E depicts genes of the WNT signaling pathway and indicates up- or down regulation in BWS cell lines and Table 3 presents all genes differentially expressed in BWS cell lines as compared to control cell lines. These results are derived from Nanostring analysis, but no confirmation is presented either by real-time PCR or by protein assessment to support the genomic results.

Author response: We used the Nanostring technology as it represents a medium-throughput platform to evaluate the mRNA abundance profiles, providing a reproducible and fully automated analyses of the samples. The robustness of this technology was already validated as reported in several papers now quoted in the manuscript (see for example: Veldman et al., Evaluating Robustness and Sensitivity of the NanoString Technologies nCounter Platform to Enable Multiplexed Gene Expression Analysis of Clinical Samples. Cancer Res. 2015 and Gentien et al., Digital Multiplexed Gene Expression Analysis of mRNA and miRNA from Routinely Processed and Stained Cytological Smears: A Proof-of-Principle Study. Acta Cytol. 2021).

The reliability of Nanostring technology relies on the ability to quantify the expression of multiple genes without amplification steps. Conversely, in qPCR technical artifacts could be introduced (Prokopec et al., Systematic evaluation of medium-throughput mRNA abundance platforms. RNA. 2013).

Considering all these aspects, we did not perform validation by means of another approach.

We highlighted the reliability of Nanostring technology in the ‘Materials and Methods - nCounter Analysis’ paragraph (page 16 line 442) with  the following sentence and the related references: “We used Nanostring technology as it represents a medium-throughput platform to evaluate mRNA abundance profiles providing reproducible and fully automated analyses of the samples. The robustness of this technology was already validated in several papers [46, 71-72]. The reliability of Nanostring technology is based on the ability to quantify the expression of multiple genes without amplification steps. Conversely, technical artifacts could be introduced in qPCR.”

Other references that highlight the robustness and the reliability of the Nanostring technology are reported below:

  1.       Theis M, Paszkowski-Rogacz M, Weisswange I, Chakraborty D, Buchholz F. Targeting Human Long Noncoding Transcripts by Endoribonuclease-Prepared siRNAs. J Biomol Screen. 2015 Sep;20(8):1018-26. doi: 10.1177/1087057115583448.
  2.       Maxfield KE, Taus PJ, Corcoran K, Wooten J, Macion J, Zhou Y, Borromeo M, Kollipara RK, Yan J, Xie Y, Xie XJ, Whitehurst AW. Comprehensive functional characterization of cancer-testis antigens defines obligate participation in multiple hallmarks of cancer. Nat Commun. 2015 Nov 16;6:8840. doi: 10.1038/ncomms9840. PMID: 26567849; PMCID: PMC4660212.
  3.       Chen L, Engel BE, Welsh EA, Yoder SJ, Brantley SG, Chen DT, Beg AA, Cao C, Kaye FJ, Haura EB, Schabath MB, Cress WD. A Sensitive NanoString-Based Assay to Score STK11 (LKB1) Pathway Disruption in Lung Adenocarcinoma. J Thorac Oncol. 2016 Jun;11(6):838-49. doi: 10.1016/j.jtho.2016.02.009
  4.       Richard AC, Lyons PA, Peters JE, et al. Comparison of gene expression microarray data with count-based RNA measurements informs microarray interpretation. BMC Genomics. 2014;15(1):649. Published 2014 Aug 4. doi:10.1186/1471-2164-15-649
  5.       Rooney C, Geh C, Williams V, et al. Characterization of FGFR1 Locus in sqNSCLC Reveals a Broad and Heterogeneous Amplicon. PLoS One. 2016;11(2):e0149628. Published 2016 Feb 23. doi:10.1371/journal.pone.0149628

No functionals test are performed to confirm that observed changes in these cell lines are the result of IGF1R downregulation, especially when considering that these cell lines are not original, but transformed.

      Author response: We didn't perform functional tests because our aim was to provide a picture of IGF2, autophagy and WNT pathways in immortalized cell lines obtained from patients. To date no data are available about the involvement of these pathways in BWS syndrome. Our study provides the first evidence that these pathways are altered in Beckwith-Wiedemann Syndrome. Undoubtedly, the observed alterations will have to be validated on  patients’ fresh samples. We stressed this concept by modifying the last sentence of the Discussion section (page 15, line 421) as follows: “This work provides only a picture of these pathways in immortalized cell lines and  the observed alterations will have to be validated on patients’ fresh samples; however this is the first evidence of the involvement of these pathways in the Beckwith-Wiedemann syndrome and may provide a rationale for future studies”. 

However, we believe that the observed  changes are associated with the syndrome, since both patients’ and controls’ cells are immortalized, reducing the effect of the immortalization process on the results. 

Reviewer 2 Report

Comments and Suggestions for Authors

The manuscript entitled “Dysfunction in IGF2R Pathway and Associated Perturbations in Autophagy and WNT Processes in Beckwith-Wiedemann Syndrome Cell Lines” by Pileggi et al. show that alterations in the IGF2 signaling pathway may be responsible for the developmental deficiencies seen in Beckwith-Wiedemann syndrome They found that alterations in the interaction between IGF2 and its receptor may reduce the activity of the AKT/GSK-3/mTOR pathway, affecting autophagy and Wnt pathway. Indeed, the Wnt signaling is one of the important pathways in various biological processes, such as embryonic development and cell differentiation The results are sound and well presented. I have only a few comments.

 Figure 3. In some cases it is observed two bands in some proteins such as P-AKT or P-ERK. What is the meaning of these bands?

 Lines 266-270. In my opinion, it should be explained in more detail why these pathways indicate that WNT is involved in these processes because it will be a key point in the conclusions of the work.

 Lines 314-317. Is it possible that IGF2 levels are key to the level of activation of its receptor and therefore to the changes that occur intracellularly?

 Page 12. The AKT/GSK-3/mTOR signaling is known to be involved in controlling the incorporation of receptors into the membrane. Therefore, a decrease in the activation of this pathway could decrease the number of IGF2 receptors in the membrane and lead to more important changes in intracellular signaling. This topic could be discussed in this paragraph.

Author Response

Figure 3. In some cases it is observed two bands in some proteins such as P-AKT or P-ERK. What is the meaning of these bands?

Author response: we thank the referee for the observation, and we apologize if the activation study by western blot of our manuscript is not clear. We used a rabbit monoclonal antibody that detects the levels of p44 and p42 MAP Kinase, ERK1 and ERK2, when it is phosphorylated at Thr202 and Tyr204 (line 204 in Results paragraph). To clarify this issue, we modified the results sentence (page 7 lines 201-204) as follows “Additionally, the expression and phosphorylation at Thr202/Tyr204 of ERK1/2 (p44 and p42 MAP Kinase)  did not exhibit variation across all LCL samples (Figures 3A and S2), although a slight difference was observed in the PathScan® Signaling Array which analyzed MEK1/2  (ERK1/2 regulator) phosphorylation at Ser221 and Ser217/221 (Figure S1).” and the corresponding legend specifying ERK1/2

Regarding P-AKT, the used monoclonal antibody detects the AKT1 phosphorylation at Ser473 (line 194). However, it also recognizes the AKT2 and AKT3 phosphorylated at the corresponding residues, likely detecting very closed double bands. 

 Lines 266-270. In my opinion, it should be explained in more detail why these pathways indicate that WNT is involved in these processes because it will be a key point in the conclusions of the work.

Author response: thank you for your useful observations that allow us to better explain our results. To clarify this issue, we modified the results paragraph (page 8 from line 260 to line 279), Figure 5, and its legend. 

Results paragraph changes: “Principal Component Analysis (PCA), an unsupervised pattern recognition analysis  allowing an easy visualization of expression differences between samples, has been  performed using the nSolver software. PCA revealed a clustered distribution of BWS patients distinct from the scattered distribution of controls, suggesting that the alteration of the WNT pathways is a common condition in the BWS (Figure 5B). 

Pathway enrichment analysis is a bioinformatic technique used to analyze gene expression data aiming at identifying altered biological pathways or networks in a set of experimental data. This analysis was performed using nSolver software and highlighted  that the most altered sub-pathways belonging to the Vantage 3DTM RNA WNT Pathways Panel in BWS cells were the canonical WNT and the transcription factor pathways (Figure 5C, left panel). These findings are further displayed in the box plots presented in Figure 5C, right panel. These results highlight the involvement of the WNT pathways in BWS pathogenesis. In particular, Figure 5D provides a schematic representation of the observed expression alterations in the three main WNT signaling pathways belonging to the Vantage 3DTM RNA WNT Pathways Panel (specific annotation of genes is reported in Table S1 and Table 3). The altered nodes of these pathways, as identified by Pathview (nSolver Advanced Analysis Software 4.0; Figure 5D) were: p53 (DEG: TP53), Frizzled (DEG: FZD2), WNT (DEG: WNT10A), GBP (DEG: FRAT1), JNK (DEGs: MAPK9 and MAPK10), BAMBI (DEG: BAMBI), DKK (DEG: DKK4) and cycD (DEG: CCND1).” 

Figure 5 changes: we ensemble panel C and D because they are part of the same analysis and modified its legend as follows: “C) Analysis of WNT panel’s sub-pathways. Left: trend plot of pathway scores vs. sample types (CTRLs and BWS). This image shows the differences of the expression of the genes belonging to the different sub-pathways of the Vantage 3DTM RNA WNT Pathways Panel between controls and BWS. Right: the analysis of the two most dysregulated sub-pathways (canonical WNT and the transcription factor) is depicted also as box plots. D) Schematic representation of DEGs in the BWS cell lines belonging to the three main WNT pathways. Pathway nodes shown in white have no genes in the Vantage 3DTM RNA WNT Pathways Panel. Pathway nodes in gray have corresponding genes in the panel, however no significant differential expression is observed. Nodes in blue and orange denote downregulation or upregulation in BWS compared to CTRLs. The nodes of the pathways that were found to be dysregulated by Pathview (nSolver Advanced Analysis Software 4.0) were: p53 (DEG: TP53), Frizzled (DEG: FZD2), WNT (DEG: WNT10A), GBP (DEG: FRAT1), JNK (DEGs: MAPK9 and MAPK10), BAMBI (DEG: BAMBI), DKK (DEG: DKK4) and cycD (DEG: CCND1).

PCA, pathway enrichment analysis, box plots, and schematic representation of DEGs were performed by nSolver software (Figures rendered by Pathview, nSolver Advanced Analysis Software 4.0)” 

 Lines 314-317. Is it possible that IGF2 levels are key to the level of activation of its receptor and therefore to the changes that occur intracellularly?

Author response: thank you for your comment. We believe that our study indicates the suggested conclusion however it cannot prove it. We highlight this concept following the reviewer’s suggestion, adding the sentence in Discussion section (page 14 line 344) : "Is it possible that the amount of IGF2 is key to the level of activation of its receptor and therefore to the changes that we observed intracellularly

Page 12. The AKT/GSK-3/mTOR signaling is known to be involved in controlling the incorporation of receptors into the membrane. Therefore, a decrease in the activation of this pathway could decrease the number of IGF2 receptors in the membrane and lead to more important changes in intracellular signaling. This topic could be discussed in this paragraph.

Author response: thank you for your useful comment. According to your suggestion, we added the sentence in the Discussion section (page 14. lines 370-374)  “The AKT/GSK-3/mTOR signaling is known to be involved in controlling the incorporation of receptors (in particular IGF1R) into the membrane [67]. Therefore, a decrease in the activation of this axis could decrease the IGF2 receptor amount in the membrane and lead to more important changes in intracellular signaling”. We also added the new reference [68] supporting this topic. 

Round 2

Reviewer 1 Report

Comments and Suggestions for Authors

I  am accepting the authors response.